# The haemodynamics of the human placenta in utero

**Neele S. Dellschaft**[1,2☯], **George Hutchinson**[1,2☯], **Simon Shah**[1,2☯], **Nia W. Jones**[3], **Chris Bradley**[1,2], **Lopa Leach**[4], **Craig Platt**[5], **Richard Bowtell**[1,2], **Penny A. Gowland**[1,2]*

**1** Sir Peter Mansfield Imaging Centre, School of Physics and Astronomy, University of Nottingham, Nottingham, United Kingdom, **2** National Institute for Health Research (NIHR) Nottingham Biomedical Research Centre, Nottingham, United Kingdom, **3** Department of Child Health, Obstetrics and Gynaecology, School of Medicine, University of Nottingham, Nottingham, United Kingdom, **4** School of Life Sciences, University of Nottingham, Nottingham, United Kingdom, **5** Nottingham University Hospitals NHS Trust and University of Nottingham, Nottingham, United Kingdom

☯ These authors contributed equally to this work.
* penny.gowland@nottingham.ac.uk

**Data Availability Statement:** Data are available from the Sir Peter Mansfield Data Access Committee for researchers who meet the criteria for access to confidential data.

## Abstract

We have used magnetic resonance imaging (MRI) to provide important new insights into the function of the human placenta in utero. We have measured slow net flow and high net oxygenation in the placenta in vivo, which are consistent with efficient delivery of oxygen from mother to fetus. Our experimental evidence substantiates previous hypotheses on the effects of spiral artery remodelling in utero and also indicates rapid venous drainage from the placenta, which is important because this outflow has been largely neglected in the past. Furthermore, beyond Braxton Hicks contractions, which involve the entire uterus, we have identified a new physiological phenomenon, the 'utero-placental pump', by which the placenta and underlying uterine wall contract independently of the rest of the uterus, expelling maternal blood from the intervillous space.

## Introduction

The human placenta has a haemochorial circulation: maternal blood emerges from the spiral arteries of the uterus to bathe villous trees containing fetal vessels (Fig 1A). This circulation is critical in fetal development, yet the patterns of maternal flow within the intervillous space (IVS) and details of venous return remain largely unknown. This circulation depends on the intrauterine environment and varies considerably between mammals, and so ideally, it needs to be investigated in humans in utero [1–3].

Efficient oxygen extraction from maternal blood requires that the speed and pattern of blood flow within the IVS maintain a concentration gradient between the maternal and fetal circulations across the entire placenta at all times [4]. Fetal flow in the exchange region is slowed by distended capillary loops in the terminal branches of the villi, and spiral artery remodelling is thought to ensure relatively low-pressure, low-velocity, high-volume maternal flow in the IVS [5]. However, the net velocity of flow across the IVS has not previously been measured or linked to local movement within the IVS.

It is generally assumed that placental blood flow is in a steady state, but recent mathematical modelling has suggested that active contraction of the stem villi (which have contractile

**Funding:** This work was funded by the Human Placenta Project NIH grant 1U01HD087202-01 (PAG, NWJ, and RB) and by the Medical Research Council [grant number MC_PC_13072] (PAG and RB). The funders had no role in study design, data collection and analysis, decision to publish, or preparation of the manuscript.

**Competing interests:** The authors have declared that no competing interests exist.

**Abbreviations:** DWI, diffusion-weighted imaging; $f_{IVIM}$, intravoxel incoherent motion fraction; HC, healthy control pregnancy; IVS, intervillous space; MRI, magnetic resonance imaging; MVM, maternal vascular malperfusion; PCA, phase contrast angiogram; PE, preeclamptic pregnancy; QSM, quantitative susceptibility mapping; ROI, region of interest.

elements along their length [6]) could assist maternal and fetal circulations in the placenta [7]. However, contractions of the placenta (independent of the rest of the uterus) have never been reported in utero.

Noninvasive imaging provides a unique opportunity to address these key unknowns. Ultrasound is a very powerful technique for guiding care in clinical obstetrics; uterine artery Doppler can measure flow in larger vessels and can assess spiral artery resistance indirectly [8] but provides limited information on movement of blood within or across the IVS [9]. Magnetic resonance imaging (MRI—glossary in S1 Table) is increasingly used to study fetal growth and development, diffusion-weighted imaging (DWI) of the utero-placental unit is established as a clinical marker of abnormally invasive placentation [10], and DWI and $T_2^*$-weighted imaging are increasingly being used to assess placental development [11–20].

Our aim was to provide new insights into blood movement and blood oxygenation in the human placenta using MRI. We have made the first measurements of the velocity of bulk movement through the placenta using phase contrast angiography (PCA) MRI and have compared this with smaller-scale incoherent movements within the IVS and vascular network of the uterine wall, assessed using DWI. We used quantitative susceptibility mapping (QSM) to assess placental oxygenation [21] and dynamic imaging to detect contractions of the placenta and its underlying uterine wall, independent of the rest of the uterus. We considered preeclamptic pregnancies (PEs) as a model of altered placental function. Taken together, these results provide detailed new insights into human placental haemodynamics.

## Results

### Flow into and through the IVS

We measured net blood flow velocity using PCA, which provides directional information by encoding net velocity of the contents of a voxel in the phase of the MRI signal [22]. Fig 2A

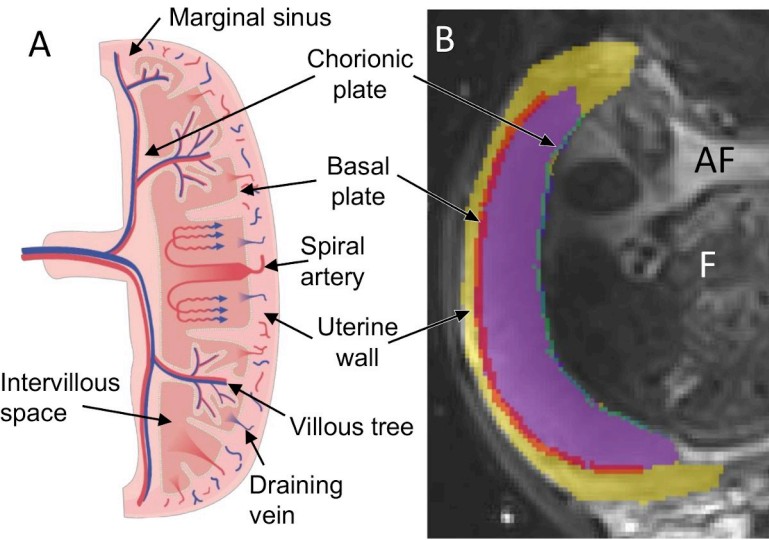

**Fig 1.** (A) Schematic showing blood movement through the placenta. Maternal blood enters the placenta via the spiral arteries before percolating through the IVS and then exiting through myometrial veins. Fetal blood flows through fetal vessels that cross the chorionic plate, before passing into stem villi (where fetal veins and arteries are closely apposed). Stem villi repeatedly branch to form the looped capillaries of the terminal villi so that maternal and fetal blood supplies are in close proximity without mixing. (B) Typical ROIs used in the analysis shaded in colour on an MRI scan (the purple area corresponds to the placental ROI); F and AF are also indicated. AF, amniotic fluid; F, fetus; IVS, intervillous space; MRI, magnetic resonance imaging; ROI, region of interest.

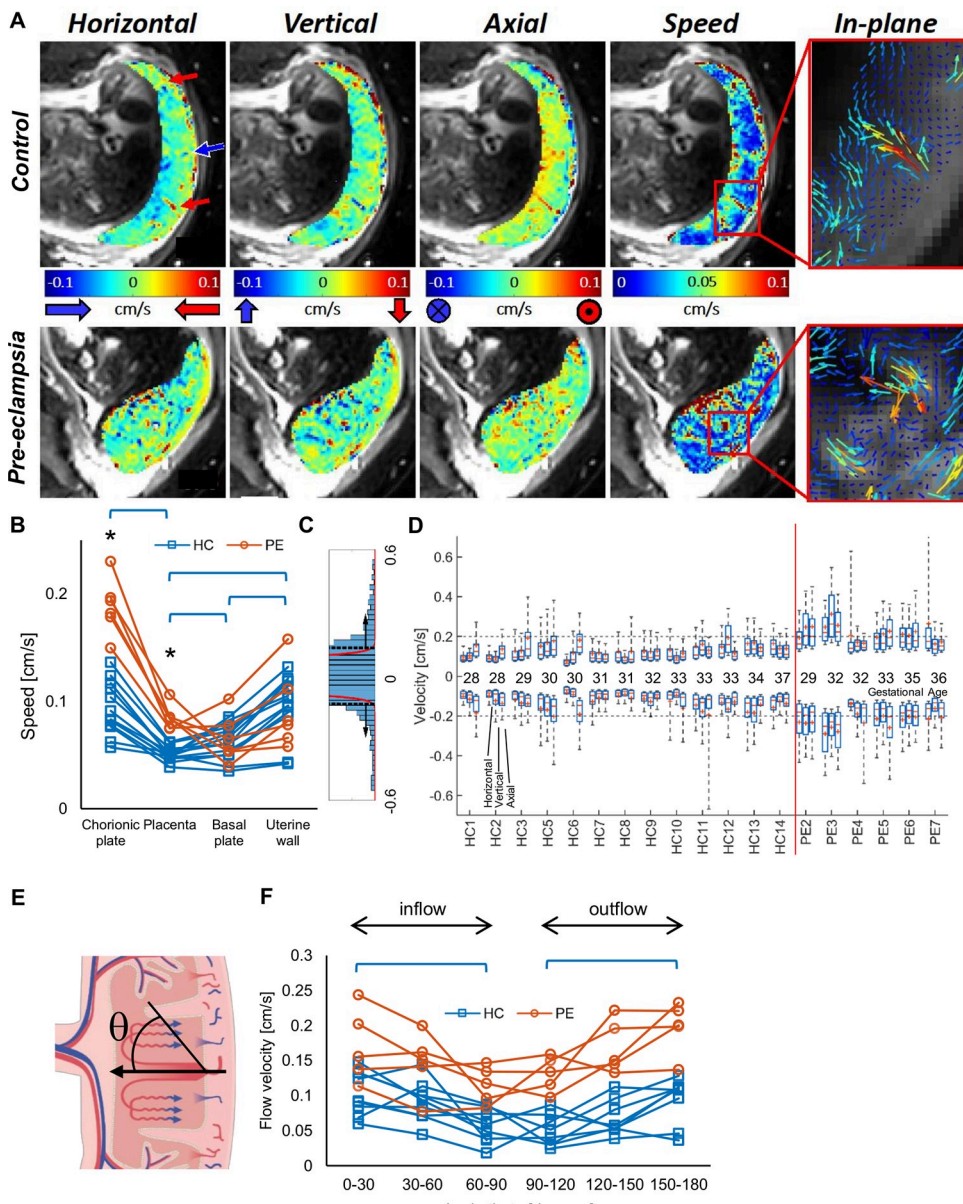

**Fig 2. Net velocity of flow through the placenta.** (A) Maps of velocity across the placenta for participants HC3 (top) and PE6 (bottom) showing velocity in three orthogonal directions, net speed (|ν|), and quiver plots that illustrate the direction and magnitude of blood flow within the imaging plane. (B) Speed of blood movement averaged across each ROI for each participant (* indicates significant difference between HC and PE; brackets indicate significant difference between ROIs for HC participants only; S3 Table). (C) The distribution of one component of velocity across the placental ROI for one HC, magnified to illustrate the tails of the histogram. The red curve indicates a normal distribution fitted to data within one standard deviation of the full histogram, and the dashed lines indicate four standard deviations from the centre of that fitted peak; voxels beyond these dashed lines were identified as fast-moving voxels. (D) The median (−), mean (+), interquartile range (box), and absolute range of the horizontal, vertical, and axial velocity in these fast-moving voxels. (E) The velocity of flow within the placenta in a narrow band adjacent to the basal plate, binned by direction of flow with respect to the normal to the basal plate, (F) showing that the highest velocities are perpendicular to the basal plate for both inflow and outflow. Underlying data plotted in panels B, D, and F are provided in S1 Data. HC, healthy control pregnancy; PE, preeclamptic pregnancy; ROI, region of interest.

shows maps of the three orthogonal components of velocity and net speed |*v*|, highlighting regularly spaced regions of relatively fast-moving blood (>0.1 cm/s), interspersed with regions of slow-moving blood (<0.05 cm/s). Quiver plots illustrate blood streaming from spiral arteries, percolating through the IVS, and returning to the uterine veins and resemble the 'fountains' predicted by mathematical models [23,24]. We often observed small regions of high velocity at the periphery of the placenta and broad regions of slow coherent movement near the chorionic plate (Fig 2A). These patterns were stable over repeated measurements in the absence of gross movement (S1 Fig). For healthy control pregnancies (HCs), the average net speed of flow was fastest in the uterine wall (see regions of interest [ROIs] in Fig 1B) and slowed progressively through the vessels of the uterine wall and basal plate into the IVS (Fig 2B, *P* = 0.002 for net speed in uterine wall versus placenta). Fig 2B also shows that as expected, the velocity was higher in the independent fetal circulation of the chorionic plate than in the placenta. We also wanted to isolate voxels in the placenta in which the net velocity was high. To do this, we plotted the histogram of the distribution of each component of velocity across the placental ROI. Fig 2C shows an example for one component, for one HC, magnified to illustrate the tails of the histogram. The red curve indicates a normal distribution fitted to data within one standard deviation of the full histogram, and the dashed lines indicate four standard deviations from the centre of that fitted peak; voxels beyond these dashed lines were identified as fast-moving voxels. Fig 2D considers only such voxels within the placenta and shows that the high velocities were similar for all three components in both forward and reverse directions for each participant. The flow measured in the placenta (<0.2 cm/s) was slower than predicted by some computational models (5 cm/s [24]) but supports others (0.6 cm/s [25]) and explains the long transit times previously measured with contrast-enhanced MRI [24] and arterial spin labelling [26]. It also confirms the hypothesis that spiral artery remodelling in early pregnancy slows blood flow as it enters the IVS [27]. Fig 2E and 2F focuses on a strip of placenta adjacent to the uterine wall and shows that flows into and out of the placenta are of similar velocity and faster than flow parallel to the basal plate for HCs (*P* = 0.015 IN, *P* = 0.050 OUT). This indicates that the speed of venous drainage, which is crucial for proper circulation in the IVS, is similar to that for arterial input.

Placentas from PEs (with or without fetal growth restriction) showed higher mean velocity (Fig 2B, S3A Table, *P* < 0.001) and a wider range of velocities (Fig 2D, S3B Table, averaging over all three components of velocity for each participant, *P* < 0.001). This is consistent with incomplete spiral artery remodelling and villous hypoplasia. Furthermore, the blood flow parallel and perpendicular to the basal plate was also faster in the PEs for which it could be measured (Fig 2E and 2F, PE faster than HC *P* < 0.05 in every angular range). The blood also moved faster in the chorionic plate in the PE group (Fig 2B, S3A Table, *P* < 0.001), which may be due to changes in resistance of the fetal villous circulation caused by changes in fetal cardiac output [28] or altered villous branching and villous tone associated with preeclampsia related to altered angiogenic factors [29,30]. Histological confirmation of the placental anomalies is included where available and confirms maternal vascular malperfusion (MVM) with villous hypoplasia and ischaemia (S1 Data).

However, placental exchange depends not just on bulk movement but also on smaller-scale movement within the IVS, which we studied with DWI. DWI is sensitive to any incoherent movement that causes variations in MRI signal phase across a voxel, leading to attenuation of the net signal, but is not sensitive to coherent net flow. Fig 3A illustrates the change in DWI signal with increasing sensitivity to incoherent motion, defined by the diffusion weighting (*b*-value, in units of s/mm$^2$). The black and green lines indicate the expected effect of diffusion in stationary free water and blood due to restriction by cells [31]. The red bi-exponential curve includes a rapidly decaying component corresponding to the fraction of the voxel volume

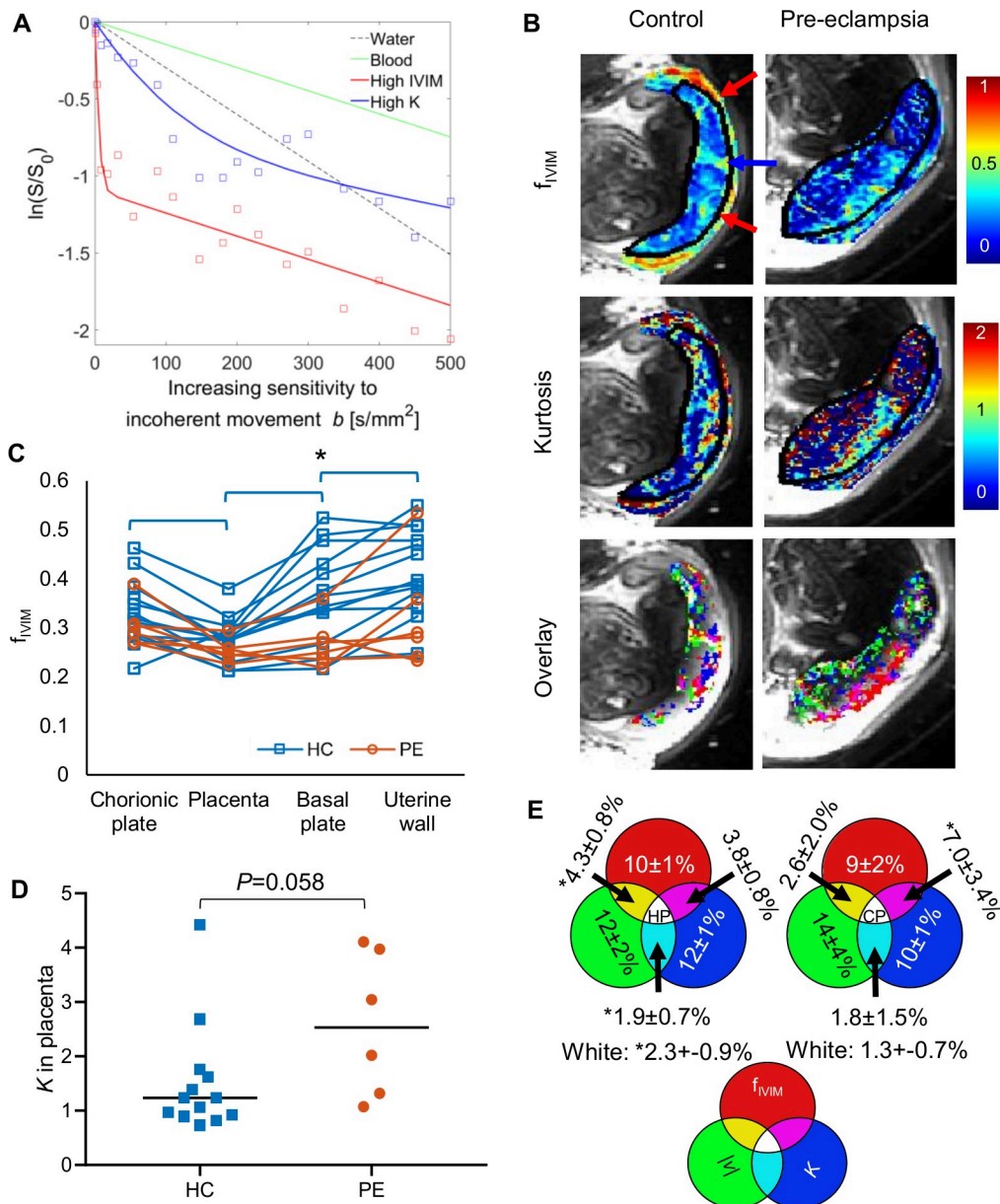

**Fig 3. Incoherent flow in the placenta and its relationship to regions of high coherent flow.** (A) Variation of signal with increasing sensitivity to incoherent movement (termed *b*-value in MRI). The black line shows the decay expected for unrestricted diffusion in stationary water. The green line illustrates slower decay typical of restricted diffusion by cells in blood samples [31]. The other curves show example placental data fitted to Eq 1. The red curve shows fast decay at low *b*-values caused by blood moving incoherently within a voxel, for instance, because of turbulence. The blue curve shows faster decay than expected due to diffusion in static blood, suggesting that flow is causing some incoherent motion in the voxel; the curvature indicates non-Gaussian diffusion consistent with percolation through a porous medium. (B) Maps of $f_{IVIM}$, kurtosis, and overlaid masks of voxels with highest 20% of values of $f_{IVIM}$, *K*, or |*v*| for HC3 and PE6. The red and blue arrows correspond to the arrows on Fig 2A. (C) Scatterplot showing average values of $f_{IVIM}$ in the chorionic plate, placenta, basal plate, and uterine wall ROIs for each participant (* indicates significant difference between HC and PE; brackets indicate significant difference between ROIs for HC participants only, S3 Table). (D) Kurtosis in the placenta region for HC and PE; the black line indicates the median. (E) Venn diagrams indicating top 20th centile of voxels in each primary ROI for both the HC and PE groups (mean ± standard deviation). *Value differs from value expected by chance by more than interparticipant standard deviation. Underlying data plotted in panels A, C, and D are provided in S1 Data. $f_{IVIM}$, IVIM fraction; HC, healthy control pregnancy; IVIM, intravoxel incoherent motion; MRI, magnetic resonance imaging; PE, preeclamptic pregnancy; ROI, region of interest.

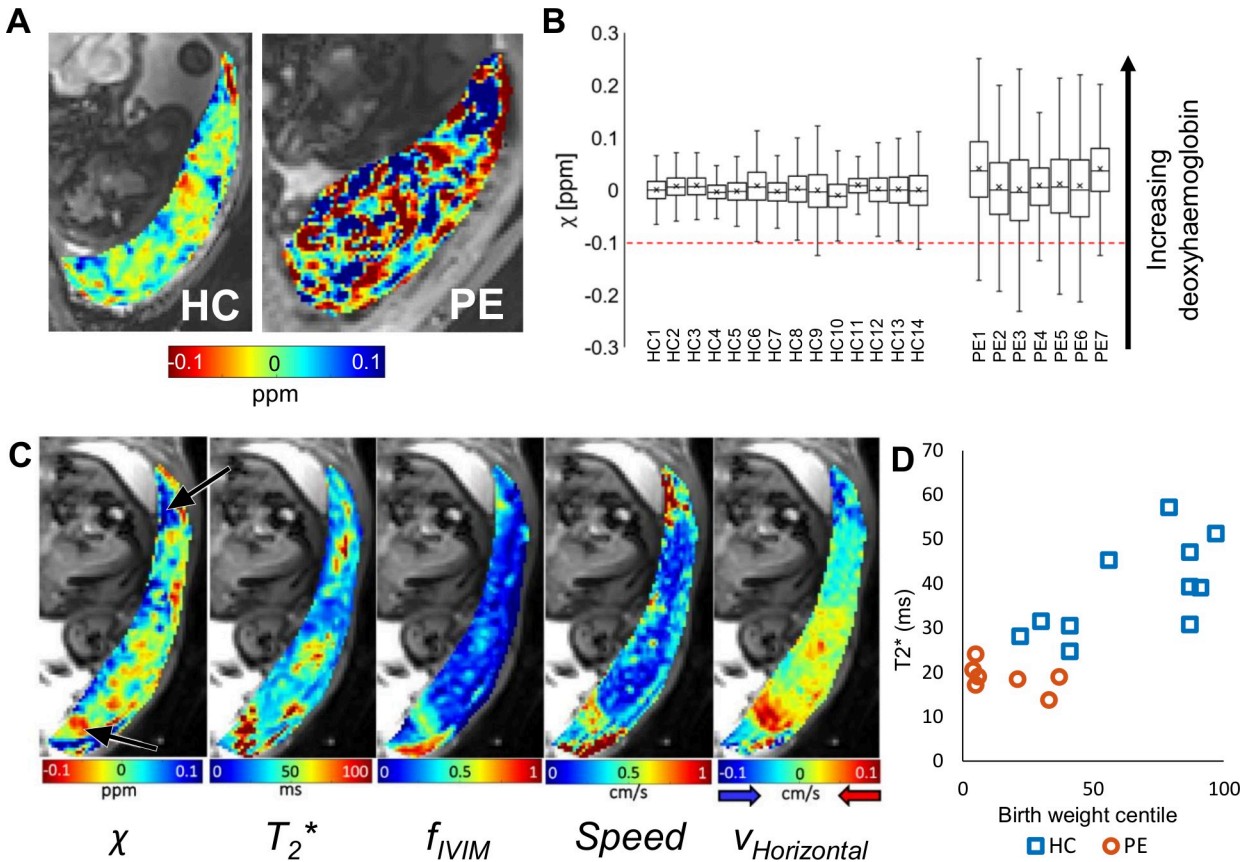

**Fig 4. Magnetic susceptibility of the placenta.** (A) Maps of magnetic susceptibility ($\chi$) of the placenta (blue corresponds to deoxygenated blood) for HC3 (moved slightly since the acquisition in Fig 2A) and PE6. (B) Plot of the median (−), mean (+), interquartile range (box), and absolute range of susceptibilities ($\chi$) for the HCs and PEs. The red dotted line indicates the value expected for fully oxygenated blood. The $\chi$ of the PE placentas is significantly higher ($P = 0.010$). (C) Maps of $\chi$, $T_2^*$, $f_{IVIM}$, net speed, and horizontal velocity for HC8. The lower arrow indicates oxygenated blood flowing into the placenta, and the upper arrow indicates deoxygenated blood flowing out of the placenta (which was also associated with a reduced $T_2^*$). (D) Variation of $T_2^*$ with birth weight centile. Underlying data plotted in panels B and D are provided in S1 Data. $f_{IVIM}$, intravoxel incoherent motion fraction; HC, healthy control pregnancy; PE, preeclamptic pregnancy.

undergoing turbulent or shearing movement in the IVS or flowing through a randomly orientated vascular network in the uterine wall and is characterised by the intravoxel incoherent motion fraction ($f_{IVIM}$ [32]). The blue line shows a nonlinear decay that is slower than for $f_{IVIM}$ (red line) but faster than for diffusion in blood (green line) and indicates random molecular displacements described by a non-Gaussian probability density function which can be characterised by kurtosis, $K$ [33]. The kurtosis we observe in the placenta is much greater than that observed for restricted diffusion in the brain. This would be expected because the blood is percolating through the IVS (rather than restricted Brownian motion, which gives smaller values of $K$ in neuroimaging). In reality, a voxel may contain many different types of motion, so we have modelled the data with this empirical equation:

$$S(b) = S_0(1 - f_{IVIM})e^{-bD+\frac{1}{6}b^2D^2K} + S_0 f_{IVIM}e^{-bD^*} \tag{1}$$

where $S_0$ is the underlying MR signal, $b$ is the diffusion weighting, $D$ is the apparent diffusion coefficient, and $D^*$ is the pseudo-diffusion coefficient associated with $f_{IVIM}$.

Maps revealed that bands of high $f_{IVIM}$ often coincided with or were adjacent to regions where fast, coherently moving blood entered the placenta (compare Figs 3B and 2A and Fig

4C). These suggest that the pressure drop between the spiral arteries and IVS remains sufficient to propel jets of blood across almost the whole placental thickness. Bands of high $f_{IVIM}$ were also observed corresponding to venous outflow from the placenta into the uterine wall. In the placenta of HCs, regions of high $K$ were generally distributed adjacent to focal regions of high $f_{IVIM}$ (Fig 3B), and $f_{IVIM}$ decreased moving from the uterine wall to the basal plate and placental body (placenta versus uterine wall: $P = 0.001$ for $f_{IVIM}$; S3A Table, Fig 3C).

In PE, $f_{IVIM}$ was reduced in the basal plate as previously reported [11,34] ($P = 0.046$ and $P = 0.058$ for basal plate and uterine wall), which could be the effect of the expected reduction in flow in those regions [27]. However, we found no significant change in $f_{IVIM}$ within the placental body in PE, probably because of the counteracting effects of increased flow speed, reduced blood volume, and altered villous density. Related literature is reviewed and discussed in S4 Table. There was a trend for $K$ to be increased in PE placentas compared with HCs ($P = 0.058$, Fig 3D), and this was dominated by PE patients also showing fetal growth restriction, suggesting altered percolation through the IVS in this group where villous structure is expected to be altered [35]. However, full interpretation of $K$ requires quantitative histological comparisons.

The velocity and DWI data sets were acquired independently, but for six HC and six PE participants, the placental data could be overlaid. Fig 3B shows the overlap of masks of the top 20th centile for values for $|v|$ (green), $f_{IVIM}$ (red), or $K$ (blue), and Fig 3E summarizes the percentage of voxels lying in the overlap regions of these masks, averaged across participants. Areas of high-speed blood flow streaming out of a spiral artery or into a vein were usually adjacent to areas of high incoherence ($f_{IVIM}$), and the area of overlap (yellow and white) was larger than expected by chance. This suggests that deceleration of fast coherent flow within the placenta drives the creation of eddies and retrograde shearing (see quiver plots in Fig 2A and fountains predicted by mathematical models [23,24]), which will be critical for mixing in the IVS. Areas of high $K$, indicating driven percolation through the villous trees, showed low net speed (0.04 cm/s HC; 0.07 cm/s PE). These were less likely than expected to overlap with areas of high flow, suggesting high-speed blood flows in cavities within the placental lobules as previously predicted [27]. The changes seen in PE suggest altered dissipation of the kinetic energy from the incoming blood, which could be used to investigate the relationship between high-pressure flow and damage to villi in preeclampsia [27].

## Placental oxygenation

Magnetic susceptibility ($\chi$) was measured to assess placental oxygenation. Fig 4A indicates that blood oxygenation is high in the IVS of HCs, since for adult blood, $\chi$ ranges from $-0.1$ ppm (fully oxygenated) to $+3.4$ ppm (fully deoxygenated [36]), with values below $-0.1$ ppm potentially indicating calcification (the syncytium has $\chi \approx 0$). The maps capture the red blush of blood entering the IVS and blue strips apparently corresponding to deoxygenated blood training through the IVS (see also S6 Fig). High blood oxygenation throughout the entire IVS is essential to maintain an oxygenation gradient between the maternal and fetal circulations. For PEs, the average placental susceptibility was higher (more deoxygenated; $P = 0.010$) but notably much more variable (Mann-Whitney, $P = 0.00002$; Fig 4B) than in HCs, suggesting that in PE, deranged patterns of flow lead to variations in oxygen extraction across the placenta.

Fig 4C compares oxygenation, flow, and $T_2^*$ maps in an individual for whom all data could be overlaid. It shows areas of highly oxygenated blood flowing into the placenta and areas of less-oxygenated blood flowing out of the placenta. It can be seen that $T_2^*$ and $\chi$ maps are similar but not identical. $\chi$ provides a more direct assessment of blood oxygenation than $T_2^*$, since $T_2^*$ is also affected by the local spatial distribution of oxygenated blood and potentially

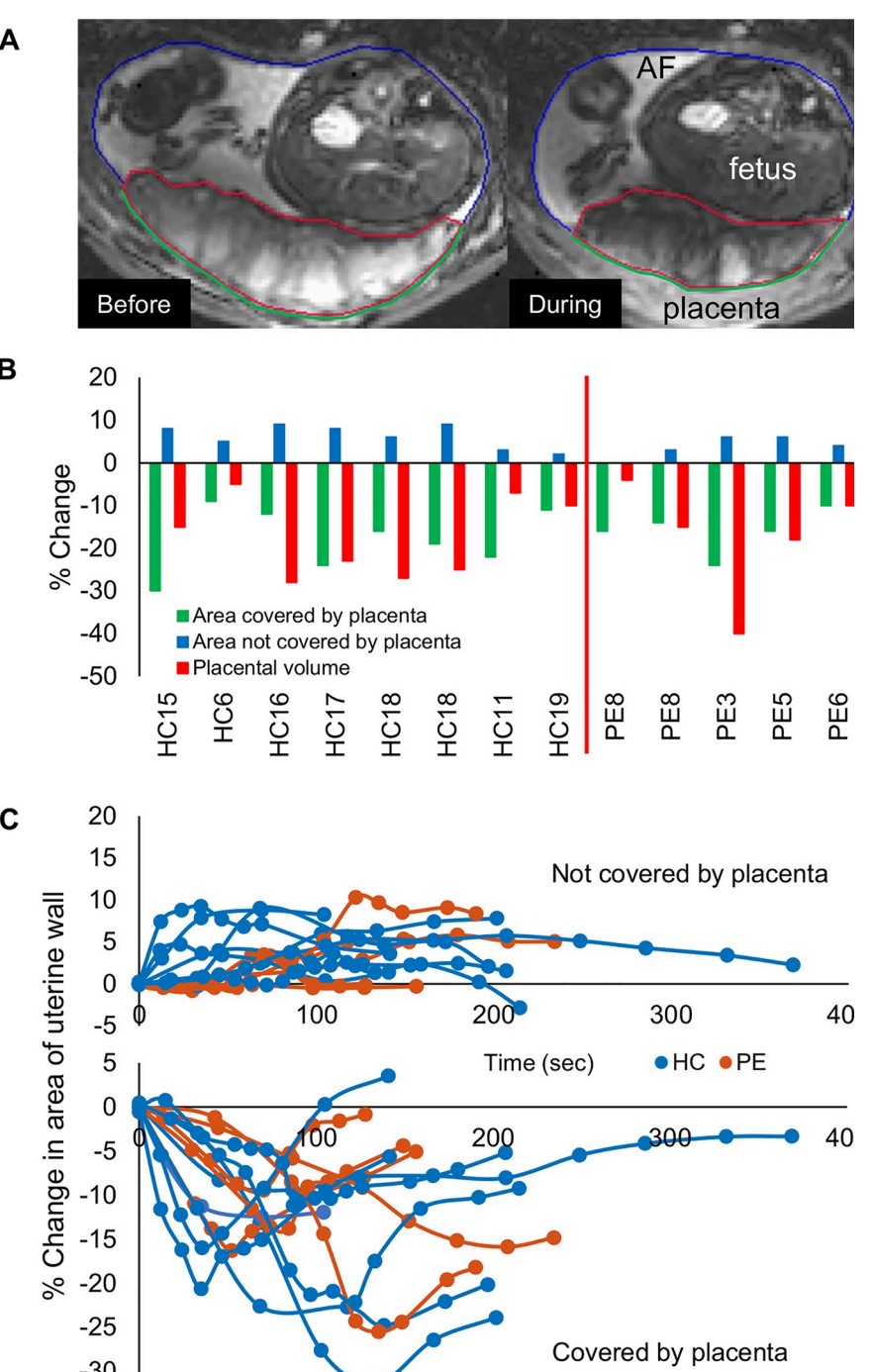

**Fig 5. The utero-placental pump.** (A) $T_2^*$-weighted images of the same placenta (HC17) before and during a contraction, with lines indicating the placental ROI (red) and the parts of the uterine wall that are covered (green) and not covered by placenta (blue). These images show a single slice, but data from slices across the whole uterus were summed to estimate the total volumes and areas involved. (B) Percentage change at maximum contraction of area of uterine wall covered by placenta, area of wall not covered by placenta, and of placental volume (green, blue, and red lines in [A], respectively), in each participant who had contractions during the 10-minute scan. Participants HC18 and PE8 had two contractions in the 10 minutes. Participants are ordered by ascending gestational age at time of scan. (C) Time course of contractions in the same participants, with each line representing the change (%) of the wall not covered by placenta (increased during contraction) and the area of the uterine wall covered by placenta (decreased during contraction). Underlying data plotted in panels B and C are provided in S1 Data. AF, amniotic fluid; HC, healthy control pregnancy; PE, preeclamptic pregnancy; ROI, region of interest.

incoherent flow. Since $T_2^*$ is sensitive to changes in both flow and oxygenation, it is expected to provide a sensitive marker of placental function. Here, we found $T_2^*$, averaged across each placenta, varied with birth weight centile even within the HC group and was significantly lower in PE than in HC, also indicating reduced and more variable oxygenation (Fig 4D, S1 Data, S3B Table; Mann-Whitney, $P = 0.00016$).

## The utero-placental pump

Subclinical uterine contractions have recently been reported in MRI [37], and we have occasionally observed these ourselves (S1 and S2 Movies). However, more frequently we have observed orchestrated contractions of just the placenta and the underlying uterine wall, leading to transient reductions in placental volume (Fig 5A and S3 and S4 Movies), with subsequent relaxation. We have termed this the 'utero-placental pump'. We observed one or more of these contractions over a 10-minute period in 12 out of 34 HCs and 7 out of 10 PEs with only three women reporting feeling any tightening when questioned immediately after the scan (S1 Data). These contractions involved a reduction in the area and thickening of the myometrium underlying (covered by) the placenta, a stretching or expansion of the rest of the uterine wall, and a simultaneous reduction in placental volume by up to 40% (Fig 5B). The contractions varied in strength and duration (Fig 5C) and caused the placenta to become thicker with less flat edges and to appear darker and more heterogeneous on $T_2^*$-weighted images, with dark bands in $T_2^*$ corresponding to bright bands on the susceptibility maps. These were different to changes seen with Braxton Hicks contractions, characterised by contraction of the entire uterine wall without alteration in placental volume (S3 Fig). With the data available, we could not detect significant differences between the contractions in HC and PEs, but to confirm this, we need to scan for longer to be able to monitor changes in the placenta before and after the contraction more frequently, possibly also using information from electrophysiology [38]. Some placental dysfunction is associated with factors that can increase placental thickness such as placental hyperinflation [39], enlargement by blood [40], chorioangiosis, and maternal or fetal anaemia. Indeed, we observed that the PE placentas were relatively thicker than the HC placentas (S1 Data and S2 Fig, $P = 0.0481$). However, we would not expect these factors to be transient or associated with changes in area of the myometrium underlying the placenta.

## Discussion

We have performed a detailed experimental investigation of maternal placental haemodynamics in utero. In healthy pregnancies, we found that the velocity of blood flow decreased from the uterine wall through to the placenta (Fig 2, S3A Table), supporting the hypothesis that spiral artery remodelling reduces the velocity of blood flow as it enters the placenta [27]. We observed relatively fast flow into and out of the placenta (Fig 2F), highlighting the importance of venous outflow, which has previously received little attention but which is crucial for proper circulation in the IVS. Around the inflow jets, we observed areas of high $f_{IVIM}$ corresponding to turbulent mixing in the IVS, and between these areas we found non-Gaussian diffusion (high kurtosis), consistent with percolation through the IVS. We found relatively uniform, high oxygenation levels across the whole placenta, consistent with previous data that have shown that the difference in oxygen partial pressure between incoming (umbilical artery) fetal blood and the IVS is low ($\approx$10 mm Hg [41] compared with $\approx$50 mm Hg in the lungs [42]), and the oxygen concentrations in the uterine artery and umbilical vein are similar [43]. These conditions of low oxygen extraction fraction suggest that relatively slow flow (low rate

of delivery [44]) is required to ensure efficient oxygen transfer from the mother to the fetus. We have measured such slow net flow in utero.

We also studied PEs as a model of altered placental function. In such pregnancies, incomplete spiral artery remodelling has been predicted to lead to higher resistance to flow and thus faster flow through the IVS [27], which we observed. We also observed that this led to increased kurtosis in the placenta and altered patterns of blood movement, probably due to the combined effects of less efficient damping of the faster inflow, and altered villous density in PE [45]. The speed of movement of blood in the chorionic plate was also faster. Importantly, susceptibility ($\chi$) measurements suggest this pattern of percolation led to lower and much more variable oxygenation across the placenta.

This study was not designed to investigate variation with gestational age, but we observed no change in flow or $f_{IVIM}$ over the second or third trimester, in agreement with some previous studies [13,46] but contrary to others [16,47,48]. Little change in $|v|$ might be expected because both placental volume and uterine artery blood flow increase with gestation.

Most importantly, we have defined a new physiological phenomenon—the 'utero-placental pump'. This causes the periodic ejection of blood from the IVS but does not affect the fetal circulation, since its entire blood volume is subject to the change in intrauterine pressure. This phenomenon may relate to previous reports that anchoring villi (connecting the chorionic plate to the uterine wall) are contractile, because such contractility would assist in maintaining the shape of the placenta as the wall contracts [7]. It may also be related to the myometrial–placental pacemaker zone that has recently been identified in rats [49]. It is likely that such utero-placental contractions may have previously been identified as Braxton Hicks contractions involving the whole uterine wall (S3C Fig), or localized uterine contractions on ultrasound [50] and MRI [50,51], which have been associated with transient reductions in both uterine blood flow [52] and placental oxygenation [37]. We agree with the previous contention that such contractions probably 'facilitate better blood flow through the placenta and the fetus' [53] and postulate that the utero-placental pump is critical in avoiding the establishment of an unstirred layer limiting exchange near the villous surface [54]. Future work is needed to assess what happens to oxygenation and blood flow before, during, and after the contraction, ideally measuring volume, oxygenation, and flow changes simultaneously [55]. Identifying an ultrasound or electrophysiological signature [38] of the 'utero-placental pump' should be a priority to accelerate research into this new phenomenon and probe its origin further. We also need to understand what triggers the contractions: is it a reduction in oxygenation or an increase in pressure within the placenta? Are the contractions initiated in the myometrium of the placental bed [49] or by the anchoring villi [7]? The rate of detection of contractions was higher in PEs than in HCs, but longer sampling in more participants is required to be confident of any difference. The shape of the healthy placenta during a contraction is similar to the thickened shape of compromised placentas [56], raising the possibility that the preeclamptic placenta is permanently contracted to some extent.

## Technical limitations

We characterised incoherent motion with a heuristic IVIM and kurtosis model [32,33], whereas the physical situation is a complex continuum, so the results will depend on the exact form of the experiment (S4 Table) and a data-led approach to analysis [55] may be more appropriate. However, the DWI sequence can also be designed to provide sensitivity to various types of motion (e.g., acceleration, circular motion, motion of different length scales [57,58]). Such data could be used to inform and test physical models that will provide better insight into IVS movement and allow us to separate the effects of speed and volume of inflowing blood and villous density. This could then be used to tune the DWI sequence to provide specific

markers of spiral artery transformation or placental porosity, which could provide new insights into placental function and new biomarkers for investigating compromised pregnancies and response to therapies.

Any dephasing due to incoherent motion causes signal attenuation in the velocity measurements, which may bias results against flows with both coherent and incoherent contributions and reduce maximum measured flows. Furthermore, the sensitivity of the velocity encoding sequence (≤0.5 cm/s) had to be decided a priori, which may have led to fast flow being underestimated (Fig 2E). In some PEs, the signal in the placenta can be very low in places because of low oxygenation, and $K$ is not meaningful if $D$ is very short because of infarcts. We excluded these regions from the analysis, which may have led us to underestimate the differences between HC and PEs.

The thickness of the placenta drops from 5.5 cm in utero [59] to 2.2 cm after delivery when much of the maternal blood is lost [60], suggesting that most of the volume in utero is made up of the maternal blood in the IVS. Therefore, we assume that MRI is mainly sensitive to blood movement in the IVS, since the voxels are too large to distinguish the counterflows in the fetal stem villi. However, sequences could be designed to separate these circulations, based on their different MR properties and scales of movement [55,61].

QSM is more directly related to tissue oxygenation than $T_2^*$, but the exact relationship is not clear. If the placenta consisted only of adult blood, a susceptibility of −0.1 ppm would correspond to fully oxygenated blood and a susceptibility of +0.3 ppm would correspond to a saturation of about 88% [36]. However, these values will overestimate the oxygenation of the maternal blood in the IVS for several reasons. Firstly, the placenta also contains villous tissue with a susceptibility of approximately zero. Secondly, fetal blood has a sensitivity to oxygenation, which is 75% of adult blood [62,63], although the fetal blood volume in the placenta is smaller and less oxygenated than the maternal blood. Compromised placentas can also include regions of haematoma containing haemoglobin degradation products (potentially more paramagnetic than deoxyhaemoglobin) and calcification (with more negative susceptibility).

Although this is a well-characterised group of women (see S1 Data demographic and Clinical information worksheets), we have no data on circulating angiogenic factors (e.g., placental growth factor) to confirm the diagnosis of preeclampsia. Histological diagnosis was possible in 50% of cases and confirmed MVM. Quantitative histological comparisons would have benefitted this study by assisting in separating different contributions to tissue magnetic susceptibility, the interpretation of $K$, and also separation of women with different forms of preeclampsia [64].

Data quality was affected by maternal and fetal movement, including uterine and placental contractions. We overacquired data to allow the selection of motion-free data sets, but alternative solutions are possible [55].

This work has provided new insights into placental blood flow and oxygenation, and the existence of the utero-placental pump. Future work should investigate the link between oxygenation and flow during contractions and effects of fetal and maternal cardiac pulsatility [9]. The results can be used to improve placental models (including in vitro models) and to optimize MRI pulse sequences to provide more specific information about placental abnormalities.

## Methods

### Ethics statement

Participants were recruited for the study from Nottingham University Hospitals NHS Trust, and written, informed consent was obtained from each participant. The study was approved by the East Midlands–Leicester Central Research Ethics Committee (16/EM/0308).

## Demographics

Eligibility criteria for the study included age 18–45 years, body mass index (BMI) $\leq$ 30 kg/m$^2$, singleton pregnancy, and no known fetal congenital anomalies. We recruited 34 women with healthy pregnancies (Control, HC) and 13 women with pregnancies compromised by placental dysfunction (preeclampsia, PE, S2 Table). In this group, preeclampsia was defined as blood pressure (BP) > 140/90 mm Hg on two separate occasions >4 hours apart with proteinuria of >30 mg/mmol on a protein–creatinine ratio or 300 mg in a 24-hour urine collection. Characteristics of the two groups are shown in S1 Data (Demographic and Clinical information worksheets). In the preeclampsia group, all women required antenatal antihypertensive treatment with labetalol, and a second agent was used in six cases. Seven of the eight women had significant proteinuria (PCR > 30 mg/mmol), and the remaining patient had a BP of 171/110 mm Hg prior to medication, raised transaminases (a diagnosis of PE based upon the International Society for the Study of Hypertension in Pregnancy [ISSHP] definition [65]) and histological confirmation of MVM. Six women with preeclampsia were, in addition, expecting a growth-restricted baby less than the 10th centile [66], all required preterm delivery, and there was one case of postnatal HELLP syndrome. Demographics were not different between HC and PE except for birth weight, birth weight centile, and gestational age at birth (S2 Table). HC16 had a placental abruption resulting in the delivery of a baby with birth weight on the 50th centile, 3 weeks after the research scan. All participants were analysed for contractions, but only a subset was analysed for blood flow in the placenta, because of changes in acquisition.

## MRI

Scanning was performed on a Philips 3T Ingenia (Best, the Netherlands), and women lay in the left or right decubitus position to avoid vena cava compression. The scanning session was split to allow the women to reposition after about 20 minutes. Posterior receive arrays that are built into the scanner bed were used along with an anterior abdominal array, positioned as close as possible to the placenta as determined from previous ultrasound, and repositioned after initial survey scans, if necessary. All scans were acquired in Normal Operating mode (whole-body averaged SAR < 2.0 W/kg). Severe RF inhomogeneity can occur in the pregnant abdomen at 3T, and parallel transmission is not currently approved for use in pregnancy on this scanner. Therefore, the optimum flip angle was determined manually as that giving maximum signal within the placenta in a single-shot gradient echo, echo-planar imaging (EPI) image (range considered 90–150$^\mathrm{o}$). All data were acquired axially to the scanner to allow rapid planning, and scanning was respiratory-gated to minimise maternal motion.

Velocity-encoded phase maps (PCAs, Fig 2) were reconstructed from the single-shot, respiratory-gated, pulsed gradient spin echo (PGSE) EPI sequence (S5 Fig, S5 Table). The PGSE sequence was also used to produce DWI maps for IVIM and kurtosis analysis in the same planes (S5 Fig and S5 Table). When considering the literature, it is important to note that the $f_{IVIM}$ and $K$ results (and to a lesser extent, $|v|$) will depend on the exact details of the sequence used (S4 Table).

For susceptibility mapping (Fig 4), gradient echo images were acquired across the whole placenta, in the same orientation as the diffusion data, using a single-shot gradient echo EPI (TE was 35 ms for HC and 20 ms for PE to provide equivalent SNR in the maps given the shorter $T_2^*$ found in PE). Multiple single-shot EPI images were acquired (TE = 20, 25, 30, and 35 ms) and fitted voxel-wise to a monoexponential decay to create $T_2^*$ maps (Fig 4C).

Finally a single-shot, respiratory-gated, $T_2^*$-weighted GRE EPI encompassing the whole uterus and placenta was acquired with respiratory gating at a minimum repetition rate of 9 seconds over a period of 10 minutes. After 2 or 4 minutes, oxygen was delivered to the mother.

We originally planned to investigate its potential as an MRI contrast agent to assess transit times, but the contractions perturbed the signals.

After scanning, women were asked to complete a questionnaire that recorded whether they experienced any painful or nonpainful contractions during the scan.

## Data analysis

No automatic motion correction was applied, because of the complexity of separate nonrigid maternal and fetal motion and the large change in contrast through the data sets. Instead, modulus and phase data sets were inspected and volumes were removed from further analysis if they showed >5-mm displacement from the reference volume due to errors in respiratory gating, or where the boundary of the placenta was deformed because of fetal movement or a contraction. If more than four data points were discarded during the low $b$-values ($b = 0$–110 s/mm$^2$) or if more than eight data points were discarded across the whole scan, then the whole data set was removed.

Additionally, for the velocity-encoded data, the unwrapped phase was assessed for every slice and encoding direction, and if the unwrapped phase was corrupted because of fetal motion or maternal respiration, that data point was discarded. Net flow speed maps were only made for slices in which data were available for all velocity encoding directions. Note: the three velocity components shown in Fig 2A were measured independently. ROIs were drawn around the placenta, uterine wall, and basal and chorionic plates on all slices on the $b = 0$ s/mm$^2$ DWI images (Fig 1B). Where possible, the same masks were then applied to the other DWI and velocity images, but if there had been movement, the ROI was redrawn, and in this case, the data sets could not be subsequently overlaid (Fig 3B and 3E).

The velocity encoding data were unwrapped by complex division of the encoded data by the unencoded data, and a high-pass Butterworth spatial filter (6 mm) was applied to data to remove global phase offsets due to respiration. Next, the blood flow velocity in each direction $v_{x,y,z}$ was calculated from:

$$v_{x,y,z} = \frac{\phi}{\gamma G_{x,y,z} T \Delta} \tag{2}$$

where $G_{x,y,z}$ is the gradient amplitude in each of the orthogonal directions, $\Delta$ is the time between the start of each gradient lobe, T is the duration of gradient lobes (S5 Fig), and $\phi$ is the value of the unwrapped filtered phase. Colour maps for $v_{x,y,z}$ (which we have called horizontal, vertical, and axial in the main text), along with the net speed maps, and quiver plots indicating the direction and amplitude of velocity projected onto the plane of the image were produced (Fig 2A). The average speed in each ROI for each participant was found (Fig 2B). Histograms of $v_{x,y,z}$ in the placenta were also created. The histograms did not show a Gaussian distribution, but their variance was calculated ($\sigma^2$) and voxels lying within $|v| < 1\sigma$ were fitted to a Gaussian distribution. Pixel values lying outside four standard deviations of this fitted distribution were then used to create box plots (Fig 2D).

In some cases, the placental signal was very low, and so a threshold was applied to avoid fitting areas with little or no signal. To determine the threshold, a close-to-zero signal region outside the participant and the mean of this ROI was taken as the noise level. Any pixel with signal less than 10 times the noise level at $b = 0$ was removed from further analysis, which was more likely to occur in PE when the $T_2^*$ was low.

To produce Fig 2F, the basal plate mask was eroded to one voxel wide and then approximated to series of lines by computing a convex hull. The angle between the in-plane velocity vector and the angle normal to the basal plate were calculated for every voxel with in-plane

speed >80th percentile, and within five voxels of these lines. The velocity was plotted against angle averaged across 30° bins (for clarity, equal angular ranges were used, despite subtending varying solid angles). One PE and one HC participant could not be included, as the shape of the basal plate was not suitable for computing a convex hull.

The DWI signals measured in each pixel at $b$-value, $S(b)$, for $b \geq 88$ s/mm$^2$ were fitted pixel-wise to:

$$S(b) = S_0[e^{-bD}] \tag{3}$$

to provide initial estimates of the apparent diffusion coefficient for water in tissue ($D$) and scaling factor $S_0$ in subsequent fits. Next, the data were fitted to the following simple IVIM model (where the terms are as in Eq 1):

$$S(b) = S_0(1 - f_{IVIM})e^{-bD} + S_0 f_{IVIM} e^{-bD^*} \tag{4}$$

The $f_{IVIM}$ component was then subtracted from the data, and the resulting data ($S'(b)$) were fitted to a model including kurtosis.

$$S\prime(b) = S_0(1 - f_{IVIM})e^{-bD + \frac{1}{6}b^2 D^2 K} \tag{5}$$

where $f_{IVIM}$ and $S_0$ are both fixed parameters using their values from the second fit (Eq 4) and $K$ is the kurtosis coefficient. The mean value in each ROI was calculated excluding voxels for which $D$ was fitted as zero ($<10^{-7}$ mm$^2$/s, so that $K$ is meaningless) or $K = 10$ (maximum value allowed by fit, often corresponding to curves that suggested refocusing of signal from regions of rapid flow at some $b$-values).

For participants for whom there was minimal movement between the independent acquisitions of the speed and $f_{IVIM}$/kurtosis data sets, pixels in the top 20th centile for speed, $f_{IVIM}$, and kurtosis were mapped (Fig 3B). The fraction of overlap between these three maps was calculated for each participant (Fig 3B and 3E).

## Quantitative susceptibility maps

Phase images from the gradient echo data were masked to the uterus (mask drawn on magnitude images) and then unwrapped using a Laplacian-based method [67] before performing a 2D V-SHARP filter [68,69] slice by slice to remove the background phase. Susceptibility maps were calculated using LSQR [21] using STI suite (https://people.eecs.berkeley.edu/~chunlei. liu/software.html; S6A Fig). The average value of susceptibility ($\chi$) within the placental ROI (on all slices) was estimated for each participant relative to the average value in an ROI of the amniotic fluid, which is assumed to have $\chi = 0$ (ROI on four slices; care was taken to avoid areas in which amniotic fluid flow from fetal movements was obvious). S6B Fig shows that the echo time (TE) has little effect on $\chi$ and that the measurement is repeatable.

## T$_2$*

Modulus images from the EPI data were masked to the placenta for two TEs (TE1 and TE2). Average signal intensities were recorded and T$_2$* calculated (Fig 4D).

## Utero-placental pump

NSD identified patients in whom she observed placental contractions during the 10-minute T$_2$*-weighted GRE EPI scan (movement within the placenta with a change in T$_2$*). Contractions were assessed by measuring the area of the entire uterine wall sectioned into (1) covered by placenta and (2) not covered by placenta, as well as total placental volume (Fig 5A and 5B).

Change of area and volume were calculated relative to stable baseline measurements before the start of the contraction. Since women were given oxygen during the acquisition to investigate its use as a method for measuring transit times, we noted whether the contraction started before or after the oxygen (S1 Data).

## Statistics

All data were compared with nonparametric tests (S2B and S3 Tables). When comparing HC and PE, we used a Mann-Whitney test (continuous data) or chi-squared test (noncontinuous data). Comparing between several ROIs, we quote the $P$ value for related-samples Friedman ANOVA (asymptotic significances) with Wilcoxon test (asymptotic, 2-tailed significances) showing a significant difference ($P < 0.05$). The overlap regions were compared with expected values for chance overlap using a test for one mean.

## Supporting information

**S1 Table. Glossary.**
(XLSX)

**S2 Table. Summary of demographic data.** Pregnant women were either classed as HC or PE. All participants were assessed for contractions, but only a subset were assessed for blood flow (velocity, $f_{IVIM}$, and $K$, due to changes in data acquisition). All data are given as median [IQR] and $P$ value for Mann-Whitney test except for parity, which is given as median [range] and $P$ value for chi-squared test. $f_{IVIM}$, intravoxel incoherent motion fraction; HC, healthy control; PE, preeclamptic pregnancy.
(XLSX)

**S3 Table.** Consolidated results and statistical outcomes of (A) $f_{IVIM}$, $K$, velocity, and $D$ in each area. (B) Average highest velocity (Fig 2D) and susceptibility. $f_{IVIM}$, intravoxel incoherent motion fraction.
(XLSX)

**S4 Table. Review of $f_{IVIM}$ measurements in the utero-placental unit in compromised pregnancies.** This table reviews all papers considering the effect of compromised pregnancy on $f_{IVIM}$ (neglecting papers studying apparent diffusion coefficient alone or twin pregnancies). $f_{IVIM}$, intravoxel incoherent motion fraction.
(XLSX)

**S5 Table. MR sequence parameters used in the acquisition of the IVIM and kurtosis, velocity, and susceptibility/contraction data.** IVIM, intravoxel incoherent motion; MR, magnetic resonance.
(XLSX)

**S1 Fig. Net speed from HC3 from three repeated scans.** The average net speed between the scans was 0.0577 ± 0.005 cm/s. HC, healthy control pregnancy.
(TIF)

**S2 Fig. Thickness-to-length ratio measured for a central slice, showing the largest area of the placenta ($P = 0.048$).** Underlying data are provided in S1 Data.
(TIF)

**S3 Fig. Utero-placental pump (HC11) compared with Braxton Hicks contraction (HC29).** Time course of contractions in the same participants as S1–S4 Movies, with each line representing the change (%) of the wall not covered by placenta (increased during contraction) and

the area of the uterine wall covered by placenta (decreased during contraction). Underlying data are provided in S1 Data. HC, healthy control pregnancy.
(TIF)

**S4 Fig. Histological examination of placenta (PE8).** Haematoxylin and eosin staining from a placenta of a patient diagnosed with severe preeclampsia and HELLP syndrome and delivered by emergency cesarean section at 28 + 6 weeks' gestational age. A low-magnification overview of the tissue (a, scale bar 5 mm) and a higher-magnification image (b, scale bar 100 μm). This image demonstrates decidual arteriopathy including focal atherosis in keeping with a histological diagnosis of maternal vascular malperfusion. Placenta examined as per 'Tissue pathway for histopathological examination of the placenta' Royal College of Pathologists, 2017. PE, pre-eclamptic pregnancy.
(PDF)

**S5 Fig. PGSE sequence diagrams.** $f_{IVIM}$ and kurtosis: Δ, the duration of gradient lobes, and $T$, the time between the start of each gradient lobe, are fixed (see S3 Table). The height of $G_{oblique}$ is increased to increase the $b$-value. After the second gradient, the magnitude of the MR signal is acquired. Velocity: Δ and $T$ have different lengths compared with the $f_{IVIM}$ and kurtosis acquisition (see S3 Table). The gradient height is set to 21 mT/m for maximum velocity encoding of ±0.5 cm/s. After the second gradient, the MR phase signal (Φ) is reconstructed for velocity measurement. $f_{IVIM}$, intravoxel incoherent motion fraction; MR, magnetic resonance; PGSE, pulsed gradient spin echo.
(TIF)

**S6 Fig. Pipeline for QSM.** (A) First column: The raw magnitude and phase data acquired from the gradient echo acquisition (S3 Table). Second column: The uterus is masked in the magnitude data, and phase images are unwrapped within that mask using a Laplacian-based method. Third column: A 2D V-SHARP filter slice by slice to remove the background phase is applied to the unwrapped phase, and subsequently the uterus mask is applied to make the filtered phase. Fourth column: Susceptibility maps are calculated using LSQR [70]. (B) The χ calculated using the QSM pipeline for three different TE values shows its general insensitivity to the TE used in the gradient echo EPI acquisition. EPI, echo-planar imaging; QSM, quantitative susceptibility mapping; TE, echo time.
(TIF)

**S1 Data. Demographic and clinical information and blood flow and contraction results.** Separate worksheets provide demographic information, clinical information for preeclamptic participants, and contraction + haemodynamic results. Clinical information includes BP, mean arterial pressure, maximum urinary protein, first/second trimester placental biomarkers, placental histology, weight (centile) and feto-placental ratio (healthy controls had BP < 135/80; two HCs had reduced PAPP-A [<0.415 MoM]), and contraction and blood flow data for each participant. Asterisk in 'Contraction', 10-minute scan was DWI instead of EPI, not allowing detailed analysis of contraction. A, aspirin; AB, abruption; AN, anaemia; BP, blood pressure; D, diabetes; DM, data missing; EDF, end diastolic flow; ElCS, elective cesarean; EmCS, emergency cesarean; EPI, echo-planar imaging; FGR, fetal growth restriction; I, insulin; I-AVD, induced labour, instrumental vaginal delivery; I-VD, induced labour, unassisted vaginal delivery; L, labetalol; M, metformin; MoM, multiple of median; MS, magnesium sulphate given at delivery; MVM, maternal vascular malperfusion; N, nifedipine; occ., women reported occasional tightenings during the scan; P, partial or potential contraction (subtle); PAPP-A, pregnancy-associated plasma protein A; PE, preeclampsia; PI, pulsatility index; rev. EDF, reversed end diastolic flow; S-AVD, spontaneous labour, instrumental vaginal delivery; S-VD,

spontaneous labour, unassisted vaginal delivery.
(XLSX)

**S1 Movie. Example time lapse videos for a Braxton Hicks contraction with markings demonstrating the movements (HC29).** HC, healthy control pregnancy.
(MP4)

**S2 Movie. Example time lapse videos for a Braxton Hicks contraction without markings demonstrating the movements (HC29).** HC, healthy control pregnancy.
(MP4)

**S3 Movie. Example time lapse videos for a utero-placental pump contraction with markings demonstrating the movements (HC11).** HC, healthy control pregnancy.
(MP4)

**S4 Movie. Example time lapse videos for a utero-placental pump contraction without markings demonstrating the movements (HC11).** HC, healthy control pregnancy.
(MP4)

## Acknowledgments

Rubie-Jo Barker produced Figs 1A and 2E.

## Author Contributions

**Conceptualization:** Nia W. Jones, Penny A. Gowland.

**Formal analysis:** Neele S. Dellschaft, George Hutchinson, Simon Shah, Nia W. Jones, Craig Platt, Penny A. Gowland.

**Funding acquisition:** Nia W. Jones, Richard Bowtell, Penny A. Gowland.

**Investigation:** Neele S. Dellschaft, George Hutchinson, Simon Shah, Nia W. Jones, Chris Bradley, Lopa Leach, Penny A. Gowland.

**Methodology:** Neele S. Dellschaft, George Hutchinson, Simon Shah, Chris Bradley, Richard Bowtell, Penny A. Gowland.

**Software:** George Hutchinson, Simon Shah, Penny A. Gowland.

**Supervision:** Penny A. Gowland.

**Writing – original draft:** Neele S. Dellschaft, Simon Shah, Nia W. Jones, Lopa Leach, Penny A. Gowland.

**Writing – review & editing:** Neele S. Dellschaft, George Hutchinson, Simon Shah, Nia W. Jones, Chris Bradley, Lopa Leach, Craig Platt, Richard Bowtell, Penny A. Gowland.

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
