## [Editor Report · Decision Letter 0]

3 Feb 2020

Dear Dr Gowland, 

Thank you for submitting your manuscript entitled "The haemodynamics of the human placenta in utero" for consideration as a Research Article by PLOS Biology.

The Academic Editor and I have now assessed your revised manuscript and I am writing to let you know that we would like to send your submission out for external peer review.

Please re-submit your manuscript within two working days, i.e. by Feb 05 2020 11:59PM.

Kind regards,

Lauren A Richardson, Ph.D

Senior Editor

PLOS Biology

---

## [Decision Letter · Decision Letter 1]

7 Mar 2020

Dear Dr Gowland,

Thank you for submitting your revised Research Article entitled "The haemodynamics of the human placenta in utero" for publication in PLOS Biology. I have now obtained advice from the original reviewers and have discussed their comments with the Academic Editor. 

Based on the reviews, we will probably accept this manuscript for publication, assuming that you will modify the manuscript to address the remaining points raised by the reviewers including the criticism from reviewer 1 regarding less sufficient histopathological correlations and improving your discussion on Figure 2C also from this reviewer. 

Please also make sure to address the data and other policy-related requests noted at the end of this email.

We expect to receive your revised manuscript within two weeks. Your revisions should address the specific points made by each reviewer. In addition to the remaining revisions and before we will be able to formally accept your manuscript and consider it "in press", we also need to ensure that your article conforms to our guidelines. A member of our team will be in touch shortly with a set of requests. As we can't proceed until these requirements are met, your swift response will help prevent delays to publication.

*Copyediting*

*Published Peer Review History*

*Early Version*

*Submitting Your Revision*

Sincerely,

Di Jiang

PLOS Biology

ETHICS STATEMENT:

-- Please include a separate subsection for the Ethics Statement in the beginning of the Methods section. 

-- Please include the full name of the IACUC/ethics committee that reviewed and approved the animal care and use protocol/permit/project license. Please also include an approval number.

-- Please include the specific national or international regulations/guidelines to which your animal care and use protocol adhered. Please note that institutional or accreditation organization guidelines (such as AAALAC) do not meet this requirement.

-- Please include information about the form of consent (written/oral) given for research involving human participants. All research involving human participants must have been approved by the authors' Institutional Review Board (IRB) or an equivalent committee, and all clinical investigation must have been conducted according to the principles expressed in the Declaration of Helsinki.

DATA POLICY:

Regardless of the method selected, please ensure that you provide the individual numerical values that underlie the summary data displayed in the following figure panels as they are essential for readers to assess your analysis and to reproduce it: 2bcdf, 3acd, 4bd, 5bc, S2, S3e. NOTE: the numerical data provided should include all replicates AND the way in which the plotted mean and errors were derived (it should not present only the mean/average values).

Reviewer remarks:

Reviewer #1: The text has improved, but there are lingering issues, largely related to the lack of histopathological correlations (except for four participants, listed in Table S2a). This was mentioned by more than one reviewer. In addition, the authors did not really address my comment regarding Fig. 2C, and in the revised version (page 4) they have removed any text reference to Fig. 2C. 

While the paper is indeed important, greater attention to readability and flow can improve the revised text. 

Reviewer #2 (Ananth Karumanchi, signed review): Manuscript is significantly improved. 

Reviewer #3 (John Kingdom, signed review): 

Thank you for taking the time to prepare thoughtful responses and edits to your manuscript, in response to all reviewers. 

I remain skeptical about your proposed concept of a utero-placental pump, but I'm open-minded enough to allow the reader to decide.

I spend a lot of time scanning pregnancies with complex placental problems, and focal (non-concentric) uterine contractions are common; they can very easily lead to mis-interpretation via placental distortion, so its important to re-assess the placenta when the uterus is soft and the myometrium is uniform. 

Some investigators have previously reported that placentas of near-term pregnancies with preeclampsia are heavier than average, and increased central placental thickness has been reported in the 1st trimester (NT scan) examination, in women who subsequently develop preeclampsia, and this phenomenon persists in the 2nd trimester. 

Most women who develop preeclampsia at term (37+ weeks) have no demonstrable placental disease ; rather, they have "maternal" disease, as opposed to "placental MVM" disease (eg see prospective cohort study, Wright E, & Kingdom J, Obstet Gynecol, Nov 2017). They have well-perfused placentas with normal uterine artery Doppler, and so do not have hypopxia-reoxygenation injury (which represses PlGF production) as is found in the preterm preeclamptic placenta, with histologic features of MVM including deciudual vasculopathy. That is why maternal PlGF testing is only valid up until 36 weeks' gestation. 

I hope you include PlGF testing and placental pathology costs in your future grant applications, as they are cost-effective and important phenotypic tests. 

Sincerely, Dr. John Kingdom MD, University of Toronto, Mount Sinai Hospital, Canada.

---

## [Editor Report · Decision Letter 2]

27 Apr 2020

Dear Dr Gowland,

On behalf of my colleagues and the Academic Editor, Masahito Ikawa, I am pleased to inform you that we will be delighted to publish your Research Article in PLOS Biology. 

Early Version

PRESS 

Kind regards,

Alice Musson

Publishing Editor, 

PLOS Biology

on behalf of

Di Jiang,

Senior Editor

PLOS Biology